# The use of the phrase "data not shown" in dental research

Eero Raittio[1], Ahmad Sofi-Mahmudi [2,3]*, Erfan Shamsoddin[2]

**1** Institute of Dentistry, University of Eastern Finland, Kuopio, Finland, **2** Cochrane Iran Associate Centre, National Institute for Medical Research Development (NIMAD), Tehran, Iran, **3** Seqiz Health Network, Kurdistan University of Medical Sciences, Seqiz, Kurdistan, Iran

* a.sofimahmudi@gmail.com, sofimahmudi@research.ac.ir

**Data Availability Statement:** The data underlying the results presented in the study are available from https://osf.io/5zryu.

**Funding:** The author(s) received no specific funding for this work.

## Abstract

### Objective

The use of phrases such as "data/results not shown" is deemed an obscure way to represent scientific findings. Our aim was to investigate how frequently papers published in dental journals use the phrases and what kind of results the authors referred to with these phrases in 2021.

### Methods

We searched the Europe PubMed Central (PMC) database for open-access articles available from studies published in PubMed-indexed dental journals until December 31st, 2021. We searched for "data/results not shown" phrases from the full texts and then calculated the proportion of articles with the phrases in all the available articles. From studies published in 2021, we evaluated whether the phrases referred to confirmatory results, negative results, peripheral results, sensitivity analysis results, future results, or other/unclear results. Journal- and publisher-related differences in publishing studies with the phrases in 2021 were tested with Fisher's exact test using the R v4.1.1 software.

### Results

The percentage of studies with the relevant phrases from the total number of studies in the database decreased from 13% to 3% between 2010 and 2020. In 2021, out of 2,434 studies published in 73 different journals by eight publishers, 67 (2.8%) used the phrases. Potential journal- and publisher-related differences in publishing studies with the phrases were detected in 2021 (p = 0.001 and p = 0.005, respectively). Most commonly, the phrases referred to negative (n = 16, 24%), peripheral (n = 22, 33%) or confirmatory (n = 11, 16%) results. The significance of unpublished results to which the phrases referred considerably varied across studies.

### Conclusion

Over the last decade, there has been a marked decrease in the use of the phrases "data/ results not shown" in dental journals. However, the phrases were still notably in use in dental studies in 2021, despite the good availability of accessible free online supplements and repositories.

**Competing interests:** The authors have declared that no competing interests exist.

## Introduction

The foundation of science and research is sustainable, valid and reliable when the results are available to be tested, replicated and reproduced [1–4]. Open access articles, data and code sharing, funding and conflicts of interest disclosures and detailed descriptions of materials, methods and results are great facilitators to open science [4–6]. However, studies suggest that frequently published results may be non-reproducible, which means the findings are difficult or impossible to reproduce [5, 7–9]. Fortunately, open science practices have also been adopted in biomedical research over the last decades [5, 10]. For instance, some journals have adopted compulsory data and code availability statements and abandoned strict word, table and figure limits. Additionally, online repositories and scientific publishers' online supplements to articles have facilitated easy and free data as well as code and document sharing. Open Science Framework (OSF, www.osf.io), figshare (www.figshare.com) and GitHub (www.github.com) are some examples of online platforms where one can manage project information and data/code sharing and archive for free.

Frequently, phrases such as "data not shown" or "results not shown" are used to refer to unpublished results. It has been assumed that that results may be related to confirmatory analyses (similar results published elsewhere), negative results, peripheral results (not directly related to the topic), sensitivity analyses or future results (e.g., results related to the manuscript in preparation) [11]. However, we are unaware of any systematically conducted study that investigated what kind of unpublished results the phrases actually refer to. Using such phrases to refer to results can be seen problematic for multiple reasons. If researchers report a considerable amount of results or important results with such phrases, and without sharing results, data or code, free interpretation and verification of results are doubly harder or even impossible [4, 5]. In other words, the use of phrases obscures transparency, reproducibility and weakens the peer review process [12]. Focusing on statistically significant results and neglecting the negative ones is a major reason behind publication bias, but may also threaten the reproducibility of scientific results [13]. In addition, proper interpretation of sensitivity analyses require that modelling modifications, parameters and sensitivity results are adequately reported [14, 15]. Results not considered important for the purposes of the current study may be crucial to conduct a systematic review or meta-analysis on the closely related topic.

However, it remains unknown how frequently published studies include such phrases to refer to unpublished results in the current era of open access online journals and free online repositories and supplements to articles. Therefore, we aimed to investigate how frequently papers published in dental journals used the phrases "data/results not shown". Accordingly, to describe current practices, we examined what kind of results the authors referred to with these phrases from the studies with the phrases published in 2021. Further, we also examined the data sharing statements, data, supplement and code availability for the studies that used these phrases.

## Materials and methods

### Protocol registration

We shared the protocol for this study on OSF on 26 September 2021 (osf.io/5zryu). All codes and data are also available on osf.io/5zryu. Deviations from the protocol are available in S1 Text.

### Bibliographic search

We conducted searches in the Europe PubMed Central (PMC) database, which contains over seven million full-text articles at the moment. First, we searched all PMC open access articles

(PMCOA) published in the PubMed-indexed dental journals in the database until 31 December 2021. Then, we searched for the phrases "data not shown" and "results not shown" from the PMCOA articles published in the same journals until the same date.

## Selection of 2021 subsample

From those searches, we selected studies published in 2021. From studies for which the search indicated that they included the phrases, we manually confirmed whether the phrase referred to unreported/unpublished results. If they did so, we included them for further analysis.

## Data extraction from the 2021 subsample

From studies published in 2021 with the "data/results not shown" phrases, we documented whether the study shared data or code or online supplementary materials/appendices within the journal website or via other platforms. Then, we categorised the studies based on whether the "data/results not shown" phrases referred to confirmatory results, negative results, peripheral results, sensitivity analysis results, future results or other/unclear categories (Table 1) [11]. We also searched for information about publishing free online supplementary materials from the websites of all journals which had published at least one paper in the subsample (yes unlimited, yes limited, unclear, no). We searched the name of the publisher of each journal from Publons (and for sensitivity analysis also from SCImago and National Library of Medicine, NLM, Catalog). All data extractions from full texts were performed first by one of the authors, and all extractions were confirmed/checked by another author; discrepancies were solved through discussion.

## Data synthesis and analyses

For each publication year, we calculated the percentage of studies with the "data/results not shown" phrases from the total number of PMCOA articles published in the same dental journals during the year. We also searched the phrases from all PubMed-indexed PMCOA articles

**Table 1. Definitions of phrase types with examples, adapted from [11].**

| Type | Definition | Example from the sample |
|---|---|---|
| Confirmatory results | Demonstrating the reproducibility of previous findings or demonstrating validity methods (e.g., negative controls). | "As elaborated in a preliminary experiment (data not shown), the amount of biofilm on the outer surface of the experimental abutments in both treatment groups was very variable." [16] |
| Negative results | Results that did not show statistically significant effect/association on some threshold, e.g., $p > 0.05$. | "The average loss of lingual ridge height between the four groups was 0.6–1.0 mm, with no significant difference among the groups ($p > 0.05$; data not shown)." [17] |
| Sensitivity analysis results | Sensitivity analyses assess the robustness of results, for instance, the impact of including or excluding some variables from the analysis. | "In addition, sensitivity analyses conducted, limiting the data to those with known values of isoprostanes and plasma carbonyls, yielded similar results (data not shown)." [18] |
| Peripheral results | Referring to results not directly relevant to the main topic of article, often mentioned in the discussion section. | "Results of different studies revealed moderate correlations between the MDAS [Modified Dental Anxiety Scale] and dentists' observations (0.4 to 0.66) [9, 34, 37]. In our study, the strength of the correlation between the dentist's observations and MDAS scoring was also moderate (results not shown)." [19] |
| Future results | Results which are going to be represented in other subsequent papers, often mentioned in the discussion. | Not in the sample |
| Other/unclear | For instance, related to statistical procedure selection (e.g., normality tests), studies with multiple phrases and varying purposes, and phrases with vague purposes. | "Faced with the intensification of these already existing barriers, the population would have a greater tendency to search the Internet for ways of self-resolution of toothache, especially in developing countries (United Nations, 2020), which presented significantly higher RSV values than developed ones (data not shown)." [20] |

and wrote down the returned number of hits for each year to make a comparison with PMCOA articles from dental journals. We reported our findings with simple descriptive tables and figures, as well as provided some examples of how the phrases were used. Journal- and publisher-related differences in the number of studies with the phrase, from the total number of PMCOA articles from each journal or publisher in 2021, were tested using Fisher's exact test with Monte Carlo simulations, with 100 000 replications. We used the R v4.1.1 software (2021-08-10, R Foundation for Statistical Computing, Vienna, Austria. http://www.R-project.org) for statistical analysis.

## Results

### Overall perspective

The search from the Europe PMC database identified 21 217 unique PMCOA articles from 116 different dental journals until 31 December 2021. Of these, the search for "data/results not shown" phrases produced 1 474 unique records from 70 various dental journals.

As the total number of PMCOA dental articles was low before 2010, there was considerable fluctuation in the percentage of studies with the phrases. As the number of available dental articles increased, the percentage of studies with the phrases stabilised to around 13% by 2010. From 2010, the percentage decreased to approximately 3% in dental journals from 2010 to 2020 (Fig 1A).

Amongst all PubMed-indexed PMCOA articles the percentages were clearly higher in all study years than in dental journals. The percentage of articles with the phrases decreased from 1990 (35%) to 2020 (6%) (Fig 1B). In both samples, these trends showed a steeper downtrend from the late 2000s onwards.

### 2021 subsample

In 2021, the search identified 2 434 unique PMCOA articles from 73 dental journals and 22 publishers. Of these, the search identified 67 PMCOA articles (from 22 different journals and eight publishers), with at least one of either of the "data/results not shown" phrases. In the full-text review, all 67 were confirmed as including the phrase(s). Thus, 2.8% of the full texts in 2021 referred to unpublished results with the phrase. Fisher's exact test showed a p-value of <0.001 for the journal-related, and 0.002 for the publisher-related differences in the percentages of studies with the phrase(s), from the total number of available PMCOA articles, from each journal or publisher in 2021 (Tables 2 and 3).

Of those 67 studies with the phrases, the phrases related most often to peripheral (n = 22, 33%), negative (n = 16, 24%) or confirmatory (n = 11, 16%) results. Few referred to the sensitivity analysis results (n = 4, 6%). However, in some cases, it was difficult to evaluate their meaning or what results the phrase referred to (n = 14, 21%). Nineteen studies used the phrase (s) multiple times (twelve studies two times, six studies three times, one study four times).

Some authors seemed to use the phrase just to indicate that certain numbers were not represented in the tables or figures but while providing data or results in the same sentence with words; for example: "The ratio of patients showing a history of head and neck cancer (19/47 vs. 14/97, $P$ = 0.0007, data not shown)..." [21].

However, in some articles, notable conclusions were made based on data not shown. For instance, in a study investigating the association of preventive dental care to healthcare outcomes, tooth extractions and endodontic treatments were given considerable attention in terms of methodological decisions, results and their interpretation, but for restorative treatments, it was just stated in the discussion that, "Furthermore, we examined the effect of receiving restorative dental care on health outcomes, but no associations were seen (data not

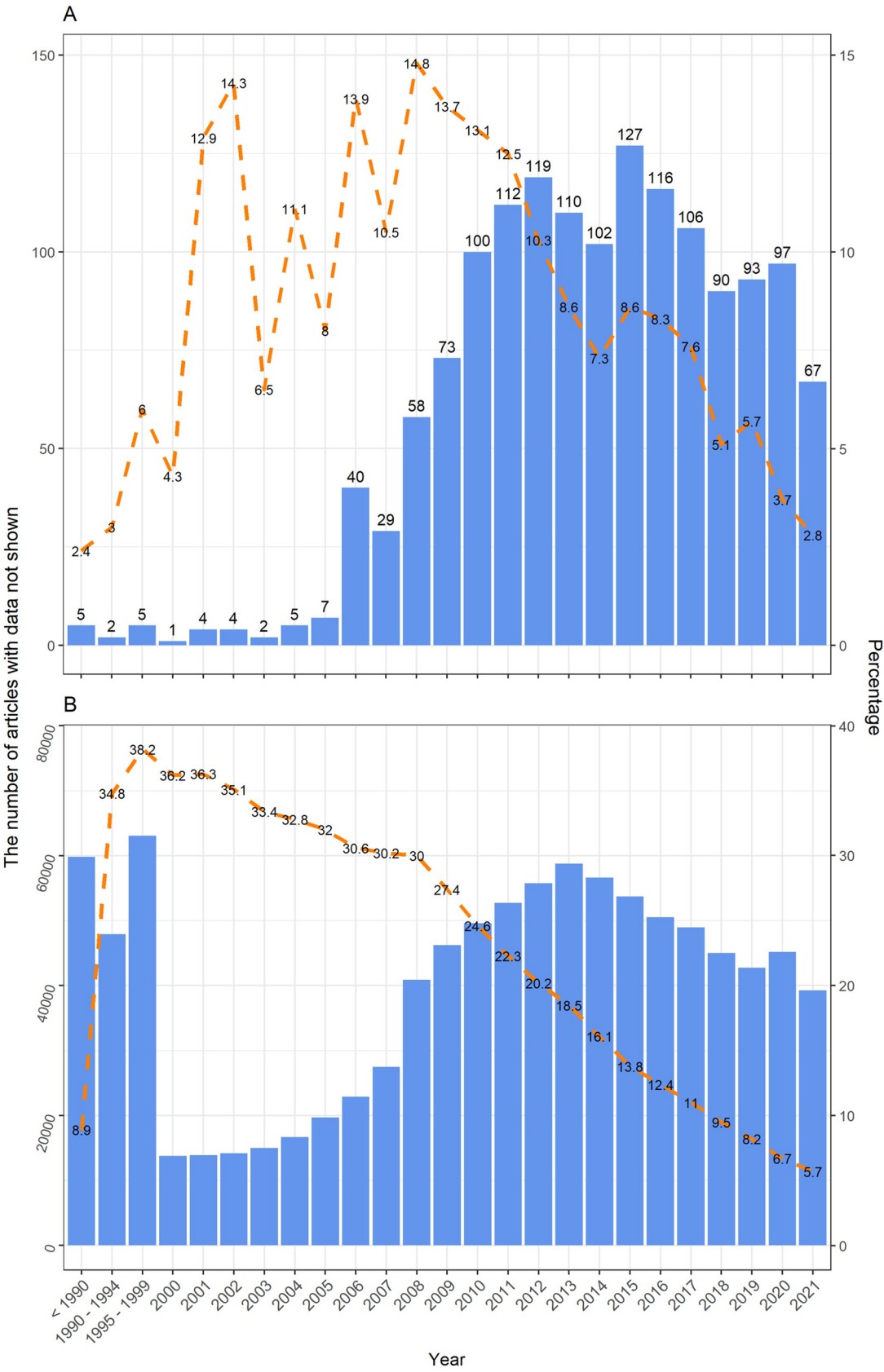

**Fig 1.** The number of available PubMed Central open access articles with "data/results not shown" phrase from dental (A) and all PubMed-indexed PMCOA (B) journals by publication year (bars), and percentage of articles with at least one "data/results not shown" phrase within each year (line).

shown)" [22]. Other examples of how the phrases were used in the studies are provided in Table 1.

Thirty-six of the studies (54%) included a data sharing statement. Three studies shared data, and no study shared code, while 27 studies (40%) included supplementary material. The search from the journal websites of 22 journals that published at least one study with the phrase(s) showed that 20 would publish online supplementary materials attached to the research articles for free, without limits. From two journals, we were unable to detect the information from their websites (The Angle Orthodontist and Medicina Oral, Patologia Oral, Cirugia Bucal). Sensitivity analysis for publisher-related differences is available in S2 Text.

## Discussion

In agreement with promising trends in open science practices in biomedical studies over the last decades [5], we found that the proportion of PMCOA articles in dental journals that used the "data/results not shown" phrases had decreased significantly, from over 10% to approximately 3%, during the last decade.

We also investigated the use of phrases in 2021 in more detail. These years researchers' have had plenty of possibilities of sharing all kinds of data via numerous free and accessible platforms. Findings showed that from all PMCOA articles published in dental journals in 2021, 67

**Table 2. The number of available open access articles from Europe PubMed Central and the number of articles with "data/results not shown" phrases in each journal.**

| Journal | Total number of studies | Studies with "data/results not shown" | Proportion (%) of studies with "data/results not shown" |
|---|---|---|---|
| BMC Oral Health | 672 | 24 | 3.5 |
| J Appl Oral Sci | 64 | 6 | 9.4 |
| Int J Oral Sci | 45 | 4 | 8.9 |
| Clin Exp Dent Res | 81 | 3 | 3.7 |
| Clin Oral Investig | 144 | 3 | 2.1 |
| J Am Dent Assoc | 41 | 3 | 7.3 |
| J Clin Periodontol | 13 | 3 | 23.1 |
| J Dent Res | 35 | 3 | 8.6 |
| J Periodontol | 4 | 3 | 75.0 |
| Head Face Med | 50 | 2 | 4.0 |
| Int J Implant Dent | 118 | 2 | 1.7 |
| Angle Orthod | 116 | 1 | 0.9 |
| Br J Oral Maxillofac Surg | 13 | 1 | 7.7 |
| Int Endod J | 2 | 1 | 50.0 |
| J Oral Pathol Med | 2 | 1 | 50.0 |
| J Periodontal Res | 6 | 1 | 16.7 |
| Mol Oral Microbiol | 2 | 1 | 50.0 |
| Odontology | 13 | 1 | 7.7 |
| Oral Dis | 33 | 1 | 3.0 |
| Pediatr Dent | 4 | 1 | 25.0 |
| Others (51 journals) | 865 | 0 | 0 |
| Total | 2434 | 67 | 2.8 |

**Table 3. The number of available open access articles from Europe PubMed Central and the number of articles with "data/results not shown" phrases in journals of each publisher.**

| Publisher | Total number of studies | Studies with "data/results not shown" | Proportion (%) of studies with "data/results not shown" |
|---|---|---|---|
| Springer Nature | 1100 | 36 | 3.3 |
| John Wiley & Sons | 294 | 15 | 5.1 |
| Faculdade de Odontologia de Bauru | 64 | 6 | 9.4 |
| American Dental Association | 41 | 3 | 7.3 |
| SAGE Publishing | 56 | 3 | 5.4 |
| Allen Press | 140 | 1 | 0.7 |
| Churchill Livingstone | 15 | 1 | 6.7 |
| Others (13 publishers) | 544 | 0 | 0 |
| Total | 2434 | 67 | 2.8 |

(2.8%) studies from 22 different journals used the "data/results not shown" phrases to refer to unpublished results. We found that there were differences in the use of the phrases between dental journals and publishers. Most commonly the phrases referred to negative, peripheral or confirmatory results. The significance of unpublished results to which the phrases referred varied considerably across studies. From the 67 studies, three studies shared raw data, and no study shared code.

Our findings showed a decreasing trend of PubMed-indexed PMCOA articles with the phrase "data/results not shown" from 1999 onwards. This trend, however, showed a steeper downtrend from 2008. This occurred after the publication of data availability editorials in Nature journals, starting from 2006 [23–25]. Thereafter, several pieces of evidence tried to elaborate and express the concerns about data availability and reproducibility of results [26–29]. In 2016, these concerns were translated into a policy of a mandatory statement on including information on whether and how others can access the underlying data for all research papers accepted for publication in Nature [30].

Some reasons for the use of the phrases to refer to unpublished results can be postulated. First, pressure to publish articles and minimising the amount of work may be one reason, that is also seen to be behind other poor scientific practices [31, 32]. In short, the results or data is not seen as worthy of publishing. Secondly, some of the results or data may be hard or impossible to share. Thirdly, as we showed, some authors just used the phrase to indicate that the results were not given in table or graphical format but were given only in text and so the results or data were not actually unpublished. However, what we see as an important reason, is that the transparency of science has not given the value in scientific practice it deserves. On the positive side, at least authors using such phrases make it honestly clear that the data or results are not shared.

Many of the studied PMCOA articles were from open access dental journals, indexed in PubMed (like BMC Oral Health and Clinical and Experimental Dental Research). Nieminen and Uribe [33] showed that in non-predatory (legitimate and indexed by established databases) open access dental journals, the presentation of results (particularly in tables and figures) was poorer than in more visible subscription-based dental journals but still better than in predatory (non-indexed) dental journals (from predatory publishers). Since referring to "data not shown" evidently is an obscure way of presenting results, it may be related to how results are presented in these studies in general.

Our findings showed that the proportion of all PubMed-indexed PMCOA articles with "data/results not shown" phrases was considerably higher than in PubMed-indexed dental journals. Whereas the decreasing trend was evident in both, all PubMed-indexed PMCOA

articles had a two-fold proportion in 2021 compared with dental journals. This potentially implies subject-related differences in the use of the phrases. So, investigating these differences in a further study could provide a better picture of the current situation.

## Implications for research policy

Solutions to enhance the movement towards open science through abandoning "data/results not shown" can be postulated. The strictest solution could be banning the use of these phrases, accompanied by editorial requests for providing the data or results not shown, as some journals and publishers have done [12]. However, a more sustainable solution could be wider adoption of open science practices, as particularly free data and code sharing have remained rare in biomedical literature over the last decades [5]. In open access journals advocating for more open science [34, 35], the obscure representation of results as well as not sharing data or code should not be overlooked by the publishers or editorial teams. For instance, during the study process, we noted a study which used the raw data availability statement template without any changes: "The data that support the findings of this study are openly available in [repository name e.g., "figshare"] at http://doi.org/[doi], reference number [reference number]" [36]. Thus it seems that despite data availability statements being mandatory in some journals, the actual content of the statement does not always receive careful consideration. Evidently, we need more commitment to open science principles from all stakeholders in science.

## Limitations

First, it is evident that the use of these phrases is not the only way to refer to unpublished results or data. Secondly, it is unknown how well PMCOA articles from dental journals represent the wider dental literature because subscription-based journals are underrepresented in the database of open access articles. However, at least in terms of transparency indicators, the differences between PubMed-indexed and PMCOA articles might be small [37]. In addition, it is worth noting that the composition of the PMCOA database varies over time due to changes in open access practices and differences in how soon after publication, the journal's articles are made available to the database (ncbi.nlm.nih.gov/pmc/journals). Further, the total number of articles included all types of papers (commentaries, letters, etc.); hence, the proportion of studies with the phrase may be higher than what was found if solely research articles had been considered. Although the analysis of PMCOA articles published in 2021 showed that the search from the database retrieved no false positives (studies without the phrases), we cannot be sure about the actual false-positive rates before 2021 or about the false-negative rate (missed studies with the phrases) of our identification strategy. Additionally, due to a large and heterogeneous sample of studies, we were unable to detect how the use of the phrases was related to other critical characteristics of studies, e.g., risks of bias. Finally, it must be noted that we do not know whether reviewers or editors had seen results to which "data not shown" referred during the peer-reviewing process which could thus justify the use of the phrase to some extent. However, editor experiences and studies have shown that (raw) data to support the findings of a study may not be shared despite reasonable requests, and sometimes given data doesn't support the conclusions made from it [38, 39].

## Conclusions

We showed that a great decrease in the use of the selected phrases occurred in PMCOA articles published in PubMed-indexed dental and other journals over the last decades. However, dental or other researchers have not completely abandoned the outdated caveat of "data/results not shown", and it was still seen to be in use in 2021. Researchers, reviewers, editorial teams

and publishers are responsible for further promoting and adopting open science practices, including providing all results, data and code, whenever possible, in a freely accessible online format, one way or another; fortunately, it is possible today.

## Supporting information

**S1 Text. Deviations from the protocol.**
(DOCX)

**S2 Text. Sensitivity analysis for publisher-related differences.**
(DOCX)

## Author Contributions

**Conceptualization:** Eero Raittio, Ahmad Sofi-Mahmudi.

**Data curation:** Eero Raittio, Ahmad Sofi-Mahmudi.

**Formal analysis:** Eero Raittio, Ahmad Sofi-Mahmudi.

**Investigation:** Eero Raittio, Ahmad Sofi-Mahmudi.

**Methodology:** Eero Raittio, Ahmad Sofi-Mahmudi.

**Project administration:** Eero Raittio, Ahmad Sofi-Mahmudi.

**Resources:** Eero Raittio, Ahmad Sofi-Mahmudi.

**Software:** Eero Raittio, Ahmad Sofi-Mahmudi.

**Supervision:** Eero Raittio, Ahmad Sofi-Mahmudi.

**Validation:** Eero Raittio, Ahmad Sofi-Mahmudi.

**Visualization:** Eero Raittio, Ahmad Sofi-Mahmudi.

**Writing – original draft:** Eero Raittio, Ahmad Sofi-Mahmudi.

**Writing – review & editing:** Eero Raittio, Ahmad Sofi-Mahmudi, Erfan Shamsoddin.

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
