## [Decision Letter · Decision Letter 0]

24 Jan 2022

PONE-D-21-34697

Movement towards open science: Have dental researchers abandoned the outdated caveat “data not shown”?

PLOS ONE

Dear Dr. Sofi-Mahmudi,

Thank you for submitting your manuscript to PLOS ONE. After careful consideration, we have decided that your manuscript does not meet our criteria for publication and must therefore be rejected.

I am sorry that we cannot be more positive on this occasion, but hope that you appreciate the reasons for this decision.

Yours sincerely,

Peter Eickholz

Academic Editor

PLOS ONE

Reviewers' comments:

Reviewer's Responses to Questions

**Comments to the Author**

1. Is the manuscript technically sound, and do the data support the conclusions?

Reviewer #1: Yes

Reviewer #2: Yes

2. Has the statistical analysis been performed appropriately and rigorously? 

Reviewer #1: Yes

Reviewer #2: I Don't Know

3. Have the authors made all data underlying the findings in their manuscript fully available?

Reviewer #1: Yes

Reviewer #2: No

4. Is the manuscript presented in an intelligible fashion and written in standard English?

Reviewer #1: Yes

Reviewer #2: Yes

5. Review Comments to the Author

Reviewer #1: The reviewer's concerns were expressed in the attached marked-up PDF document. The authors are required to better justify the necessity of this research, given the actual possibilities of a reviewer to require all available data/results, when needed.

Reviewer #2: Dear authors,

The Manuscript ID PONE-D-21-34697 entitled " Movement towards open science: Have dental researchers abandoned the outdated caveat “data not shown”?" analyses the evolution in time of the use of the phrases “data/results not shown” and to what kind of results the authors referred to with these phrases in 2021.

Although the paper is beautifully written and has few changes to be made, I am afraid I don´t find the interest in the topic. The conclusions reached add little innovation and, although I think editors should really handle this information, I don´t know if readers will get much from it.

Specifically, there are some mistakes in the text that should be corrected:

- What do authors mean by “open access”? Some of the indexed journals included in the sample are not available in open access, only subscription-based; I am afraid I don´t really understand how you made the selection of journals and editors.

- Page 4 line 44: There is a missing point after the (1-4)

- Page 6 line 98: the citation is badly spelled “m [11)”.

- The authors establish a comparison between their results from 2021 and previous data. Nevertheless, I don´t get to find in the text the original source from where you achieved previous figures. Allusions to the trends in the past are made and cited in the Discussion, but not in the Results, where the authors first make the comparison.

- Page 11 line 160: the use of the adverb “Interestingly” reads sarcastic and complicates the understanding of the statement.

- The quality of the figures is quite poor; I don´t know if it would be possible to upload them in higher resolution.

6. PLOS authors have the option to publish the peer review history of their article (what does this mean?). If published, this will include your full peer review and any attached files.

Reviewer #1: No

Reviewer #2: No

- - - - -

---

## [Author Response · Author response to Decision Letter 0]

26 Mar 2022

Authors’ response: We thank the reviewers for their constructive and insightful comments. We see that the main reason for the rejection was that the reviewers didn’t consider the study worth conducting or publishing. We definitely agree with the reviewers that the importance of the study wasn’t well communicated in the submitted version. 

First, what our study shows is what kinds of results the phrase “data not shown” refer to in the studies these days. We are unaware of any systematically conducted study that investigated what kind of unpublished results the phrases actually refer to. We see the investigation we made important for four reasons:

1) If researchers report a considerable amount of results or important results with such phrases, and without sharing results, data or code, free interpretation and verification of results are doubly harder or even impossible. In other words, the use of phrases obscures the transparency, reproducibility and weakens the peer review process.

2) As we showed, these phrases referred many times to non-statistically significant (negative) results. Focusing on statistically significant results and neglecting the negative ones is a major reason behind publication bias, but may also threaten the reproducibility of scientific results as argued by Amrhein et al (2017).

Amrhein V, Korner-Nievergelt F, Roth T. The earth is flat ( p > 0.05): significance thresholds and the crisis of unreplicable research. PeerJ. 2017 Jul 7;5:e3544.

3) Proper interpretation of sensitivity analyses require that modelling modifications, parameters and sensitivity results are adequately reported. As we showed, few papers referred to these kinds of results with the phrases.

Lash TL, Fox MP, MacLehose RF, Maldonado G, McCandless LC, Greenland S. Good practices for quantitative bias analysis. Int J Epidemiol. 2014 Dec;43(6):1969-85.

4) Results not considered important for the purposes of the current study may be crucial to conduct a systematic review or meta-analyses on the closely related topic. As we showed, many papers referred to peripheral (not directly related to the topic of study) or negative results with the phrases.

We have revised the introduction and discussion accordingly. We also made most of the revisions the reviewers suggested. We also made our analyses to cover the whole 2021 instead of including only part of 2021.

Comments to the Author

1. Is the manuscript technically sound, and do the data support the conclusions?

Reviewer #1: Yes

Reviewer #2: Yes

2. Has the statistical analysis been performed appropriately and rigorously?

Reviewer #1: Yes

Reviewer #2: I Don't Know

3. Have the authors made all data underlying the findings in their manuscript fully available?

Reviewer #1: Yes

Reviewer #2: No

Authors’ response: All data and code produced and analyzed during the study are shared via Open Science Framework https://osf.io/5zryu/

4. Is the manuscript presented in an intelligible fashion and written in standard English?

Reviewer #1: Yes

Reviewer #2: Yes

5. Review Comments to the Author

Reviewer #1: The reviewer's concerns were expressed in the attached marked-up PDF document. The authors are required to better justify the necessity of this research, given the actual possibilities of a reviewer to require all available data/results, when needed.

[We extracted the comments from the PDF:]

Although originating in the text of an authorless note (editorial ?) in Nat Chem Biol, the title seems pretentious and bombastic. This is a study of scientometrics, not a chapter of a mystery novel. Besides, the meaning is not clear. As far as the reviewer knows, a "caveat" means "warning", "caution". As such, a caveat cannot be characterized neither as "fashionable", nor as "outdated". Even more, the results of the research show clearly that the mention "data not shown" seems not to be outdated, but rather reduced as frequency of use. Please try to modify the title as to reflect the content of the research.

Authors’ response: We edited the title.

The phrase is unclear. Please reformulate.

Authors’ response: The sentence was reformulated.

This source-phrase sounds pretentious, and, at a closer look, is trivial. Hidden data always affect the transparency of a research and weakens the peer review.

Authors’ response: We reformulated the sentence.

Please justify the necessity of such a research. At what extent so far has actually the phrase "data/results not shown" produced damages in terms of interpreting the results and drawing sound conclusions? It si still not clear why this research was needed.

Authors’ response: We agree with the reviewer that the importance of the study wasn’t elaborated clearly enough. Please see the beginning of the response,

OSF means...?

Authors’ response: Open Science Framework, it was stated in the introduction.

Please explain why 2021 was chosen as a year of reference?

Authors’ response: Most recent year. Current use of the phrase is most interesting because every day sharing data and other material becomes easier and easier.

How does apply this statement to the present research?

Authors’ response: The sentence was removed.

In the Discussion section, authors should investigate the significance of the mention "data not shown" in the context of an undeterrable request for data from the reviewers. In other words, how effective could be the mention "data/results not shown", knowing that these data/results can be always requested by the reviewers?

Authors’ response: This is now mentioned in the limitations section.

Based on the distribution of the hidden data/results, in the Discussion section, authors should further speculate what could be the real reason(s) for not showing the data/results.

Authors’ response: We added a paragraph about the reasons for the use of the phrase.

Since discussing the reasons: the reviewer's experience shows that in many cases less relevant data are poorly presented/not shown, and this is not even stated as such. Given this reality, could the mention "data/results not shown" be seen as a sign of ultimate, twisted honesty from the part of the authors?

Authors’ response: This is now stated in the paragraph we added to the discussion.

It is not clear what "false-positive" or "false-negative" means in this context. Please explain.

Authors’ response: This was clarified.

Reviewer #2: Dear authors,

The Manuscript ID PONE-D-21-34697 entitled " Movement towards open science: Have dental researchers abandoned the outdated caveat “data not shown”?" analyses the evolution in time of the use of the phrases “data/results not shown” and to what kind of results the authors referred to with these phrases in 2021.

Although the paper is beautifully written and has few changes to be made, I am afraid I don´t find the interest in the topic. The conclusions reached add little innovation and, although I think editors should really handle this information, I don´t know if readers will get much from it.

Authors’ response: We agree with the reviewer that the importance of the study wasn’t elaborated clearly enough. Please see the beginning of the response,

Specifically, there are some mistakes in the text that should be corrected:

- What do authors mean by “open access”? Some of the indexed journals included in the sample are not available in open access, only subscription-based; I am afraid I don´t really understand how you made the selection of journals and editors.

Authors’ response: All PubMed-indexed dental journals were selected based on a list provided by National Library of Medicine (NLM). Open access in this context means that the article is freely accessible via Europe PubMed Central database. Therefore, the study sample also includes studies from subscription-based journals.

- Page 4 line 44: There is a missing point after the (1-4)

- Page 6 line 98: the citation is badly spelled “m [11)”.

- The authors establish a comparison between their results from 2021 and previous data. Nevertheless, I don´t get to find in the text the original source from where you achieved previous figures. Allusions to the trends in the past are made and cited in the Discussion, but not in the Results, where the authors first make the comparison.

Authors’ response: These were corrected to manuscript. 

- Page 11 line 160: the use of the adverb “Interestingly” reads sarcastic and complicates the understanding of the statement.

Authors’ response: This was removed.

- The quality of the figures is quite poor; I don´t know if it would be possible to upload them in higher resolution.

Authors’ response: Figures were improved.

---

## [Decision Letter · Decision Letter 1]

6 Jun 2022

PONE-D-21-34697R1A systematic review on the use of the phrase “data not shown” in dental researchPLOS ONE

Dear Dr. Sofi-Mahmudi,

Thank you for submitting your manuscript to PLOS ONE. After careful consideration, we feel that it has merit but does not fully meet PLOS ONE’s publication criteria as it currently stands. Therefore, we invite you to submit a revised version of the manuscript that addresses the points raised during the review process.

ACADEMIC EDITOR:

The comments of reviewer 3 should be considered to improve the article. 

My main concern is about including only studies published in 2021. Authors should highlight in the discussion why this decision and what is the impact of that. 

We look forward to receiving your revised manuscript.

Kind regards,

Rafael Sarkis-Onofre

Academic Editor

PLOS ONE

Additional Editor Comments (if provided):

Academic Editor:

The comments of reviewer 3 should be considered to improve the article.

My main concern is about including only studies published in 2021. Authors should highlight in the discussion why this decision and what is the impact of that.

Reviewers' comments:

Reviewer's Responses to Questions

**Comments to the Author**

1. If the authors have adequately addressed your comments raised in a previous round of review and you feel that this manuscript is now acceptable for publication, you may indicate that here to bypass the “Comments to the Author” section, enter your conflict of interest statement in the “Confidential to Editor” section, and submit your "Accept" recommendation.

Reviewer #3: (No Response)

2. Is the manuscript technically sound, and do the data support the conclusions?

Reviewer #3: Yes

3. Has the statistical analysis been performed appropriately and rigorously? 

Reviewer #3: Yes

4. Have the authors made all data underlying the findings in their manuscript fully available?

Reviewer #3: Yes

5. Is the manuscript presented in an intelligible fashion and written in standard English?

Reviewer #3: Yes

6. Review Comments to the Author

Reviewer #3: This is a well-written paper that addressed how frequently papers published in dental journals use the phrases “data/results not shown" and what kind of results the authors referred to with these phrases in 2021. Some study limitations cannot be disregarded, especially the underrepresentation of subscription-based journals and the lack of information on the true reasons for using such phrases. Despite that, I acknowledge that the exploratory nature of these findings can give us insightful information regarding open access practices in dental journals.

The authors have substantially improved the first version of the paper. However, I strongly suggest authors change the title. Surely the authors made a systematic search. However, this was not a systematic review, but a scientometric analysis. Also, the authors should explain why the protocol for this study was not prospectively registered.

Another comments:

Introduction, last paragraph: “Accordingly, from relevant studies in 2021, we examined….” Please consider replacing the word relevant with eligible.

Material and methods: the topic “study selection” should be rewritten and divided into three topics: “protocol registration”, “bibliographic search”, and “eligibility criteria” (or similar topics). Information regarding these three topics is all mixed in “study selection”, and the reader may find it difficult to understand.

Results: “The percentage of studies with the phrases from all the PMCOA articles decreased from 13% to approximately 3% in dental journals from 2010 to 2020 (Figure 1A)”. It is not clear why the years 2010 and 2020 were chosen.

In S4 Text (Sensitivity analysis for publisher-related differences), please provide the raw numbers and not only the p values.

7. PLOS authors have the option to publish the peer review history of their article (what does this mean?). If published, this will include your full peer review and any attached files.

Reviewer #3: No

---

## [Author Response · Author response to Decision Letter 1]

11 Jun 2022

We thank the editor and reviewers for the opportunity to submit our revised manuscript.

The comments of reviewer 3 should be considered to improve the article. 

My main concern is about including only studies published in 2021. Authors should highlight in the discussion why this decision and what is the impact of that.

Response: Our aim was to investigate reasons for current use of the phrases, because nowadays researchers have many options to share data and results freely and easily. We didn’t see it worthwhile investigating the use of phrases in more detail from the whole sample, because to some extent the use of phrases has been acceptable (for instance before widespread use of the internet).

Reviewer #3: This is a well-written paper that addressed how frequently papers published in dental journals use the phrases “data/results not shown" and what kind of results the authors referred to with these phrases in 2021. Some study limitations cannot be disregarded, especially the underrepresentation of subscription-based journals and the lack of information on the true reasons for using such phrases. Despite that, I acknowledge that the exploratory nature of these findings can give us insightful information regarding open access practices in dental journals.

Response: We thank the reviewer for his/her insightful comments.

The authors have substantially improved the first version of the paper. However, I strongly suggest authors change the title. Surely the authors made a systematic search. However, this was not a systematic review, but a scientometric analysis.

Response: We thank the reviewer for this good comment. We removed the type of study from the title completely: The use of the phrase “data not shown” in dental research

 Also, the authors should explain why the protocol for this study was not prospectively registered.

Response: The reviewer is right that the protocol was not registered beforehand, it was just shared publicly beforehand. That was our fault, we deemed it acceptable to just publish it beforehand instead of using registers such as those available via Open Science Framework. However, this doesn’t change the fact that we have worked closely following the open science principles when conducting this study.

Another comments:

Introduction, last paragraph: “Accordingly, from relevant studies in 2021, we examined….” Please consider replacing the word relevant with eligible.

Response: We edited the sentence in this regard, but also to highlight why we conducted in more detail analysis of those articles that used the phrases in 2021.

Material and methods: the topic “study selection” should be rewritten and divided into three topics: “protocol registration”, “bibliographic search”, and “eligibility criteria” (or similar topics). Information regarding these three topics is all mixed in “study selection”, and the reader may find it difficult to understand.

Response: We thank the reviewer for pointing this out. We edited and added more subsections with more detail subheaders: “protocol registration”, “bibliographic search”, “Selection of 2021 subsample”, “Data extraction from the 2021 subsample” and “Data synthesis and analyses“)

Results: “The percentage of studies with the phrases from all the PMCOA articles decreased from 13% to approximately 3% in dental journals from 2010 to 2020 (Figure 1A)”. It is not clear why the years 2010 and 2020 were chosen.

Response: We thank the reviewer for pointing this out. We improved the whole paragraph and added a more detailed description of our findings:

“As the total number of PMCOA dental articles was low before 2010, there was considerable fluctuation in the percentage of studies with the phrases. As the number of available dental articles increased, the percentage of studies with the phrases stabilised to around 13% by 2010. From 2010, the percentage decreased to approximately 3% in dental journals from 2010 to 2020 (Figure 1A). 

Amongst all PubMed-indexed PMCOA articles the percentages were clearly higher in all study years than in dental journals. The percentage of articles with the phrases decreased from 1990 (35%) to 2020 (6%) (Figure 1B). In both samples, these trends showed a steeper downtrend from the late 2000s onwards.”

In S4 Text (Sensitivity analysis for publisher-related differences), please provide the raw numbers and not only the p values.

Response: We thank the reviewer for pointing this out. Raw numbers for journal- and publisher-related differences are now provided in supplementary material available at https://osf.io/5zryu/.

---

## [Decision Letter · Decision Letter 2]

26 Jul 2022

The use of the phrase “data not shown” in dental research

PONE-D-21-34697R2

Dear Dr. Sofi-Mahmudi,

We’re pleased to inform you that your manuscript has been judged scientifically suitable for publication and will be formally accepted for publication once it meets all outstanding technical requirements.

Kind regards,

Rafael Sarkis-Onofre

Academic Editor

PLOS ONE

Additional Editor Comments (optional):

Reviewers' comments:

Reviewer's Responses to Questions

**Comments to the Author**

1. If the authors have adequately addressed your comments raised in a previous round of review and you feel that this manuscript is now acceptable for publication, you may indicate that here to bypass the “Comments to the Author” section, enter your conflict of interest statement in the “Confidential to Editor” section, and submit your "Accept" recommendation.

Reviewer #3: All comments have been addressed

2. Is the manuscript technically sound, and do the data support the conclusions?

Reviewer #3: Yes

3. Has the statistical analysis been performed appropriately and rigorously? 

Reviewer #3: Yes

4. Have the authors made all data underlying the findings in their manuscript fully available?

Reviewer #3: Yes

5. Is the manuscript presented in an intelligible fashion and written in standard English?

Reviewer #3: Yes

6. Review Comments to the Author

Reviewer #3: The authors have adequately addressed all the issues previoulsy raised and therefore I recommend publication.

7. PLOS authors have the option to publish the peer review history of their article (what does this mean?). If published, this will include your full peer review and any attached files.

Reviewer #3: No

---

## [Editor Report · Acceptance letter]

28 Jul 2022

PONE-D-21-34697R2 

The use of the phrase “data not shown” in dental research 

Dear Dr. Sofi-Mahmudi:

I'm pleased to inform you that your manuscript has been deemed suitable for publication in PLOS ONE. Congratulations! Your manuscript is now with our production department. 

Kind regards, 

on behalf of

Dr. Rafael Sarkis-Onofre 

Academic Editor

PLOS ONE